# Somatostatin Receptor Gene Functions in Growth Regulation in Bivalve Scallop and Clam

**DOI:** 10.3390/ijms25094813

**Published:** 2024-04-28

**Authors:** Xiangchao Zhang, Yuli Niu, Can Gao, Lingling Kong, Zujing Yang, Lirong Chang, Xiangfu Kong, Zhenmin Bao, Xiaoli Hu

**Affiliations:** 1MOE Key Laboratory of Marine Genetics and Breeding, College of Marine Life Sciences, Ocean University of China, Qingdao 266003, China; zhangxiangchao@ouc.edu.cn (X.Z.); nly0321@163.com (Y.N.); gaocan13934274533@163.com (C.G.); kongll_ouc@126.com (L.K.); yzj@ouc.edu.cn (Z.Y.); changlirong@stu.ouc.edu.cn (L.C.); 15621456335@163.com (X.K.); zmbao@ouc.edu.cn (Z.B.); 2Laboratory for Marine Fisheries Science and Food Production Processes, Qingdao National Laboratory for Marine Science and Technology, Qingdao 266237, China; 3Key Laboratory of Tropical Aquatic Germplasm of Hainan Province, Sanya Oceanographic Institution, Ocean University of China, Sanya 572000, China

**Keywords:** somatostatin receptor, growth regulation, *Patinopecten yessoensis*, *Mulinia lateralis*

## Abstract

Bivalves hold an important role in marine aquaculture and the identification of growth-related genes in bivalves could contribute to a better understanding of the mechanism governing their growth, which may benefit high-yielding bivalve breeding. Somatostatin receptor (SSTR) is a conserved negative regulator of growth in vertebrates. Although *SSTR* genes have been identified in invertebrates, their involvement in growth regulation remains unclear. Here, we identified seven *SSTR*s (*PySSTR*s) in the Yesso scallop, *Patinopecten yessoensis*, which is an economically important bivalve cultured in East Asia. Among the three *PySSTR*s (*PySSTR-1*, *-2*, and *-3*) expressed in adult tissues, *PySSTR-1* showed significantly lower expression in fast-growing scallops than in slow-growing scallops. Then, the function of this gene in growth regulation was evaluated in dwarf surf clams (*Mulinia lateralis*), a potential model bivalve cultured in the lab, via RNA interference (RNAi) through feeding the clams *Escherichia coli* containing plasmids expressing double-stranded RNAs (dsRNAs) targeting *MlSSTR-1*. Suppressing the expression of *MlSSTR-1*, the homolog of *PySSTR-1* in *M. lateralis*, resulted in a significant increase in shell length, shell width, shell height, soft tissue weight, and muscle weight by 20%, 22%, 20%, 79%, and 92%, respectively. A transcriptome analysis indicated that the up-regulated genes after *MlSSTR-1* expression inhibition were significantly enriched in the fat digestion and absorption pathway and the insulin pathway. In summary, we systemically identified the *SSTR gene*s in *P. yessoensis* and revealed the growth-inhibitory role of *SSTR-1* in bivalves. This study indicates the conserved function of somatostatin signaling in growth regulation, and ingesting dsRNA-expressing bacteria is a useful way to verify gene function in bivalves. *SSTR-1* is a candidate target for gene editing in bivalves to promote growth and could be used in the breeding of fast-growing bivalves.

## 1. Introduction

Bivalve culture accounted for over half of the maricultural production in China. Increasing farming harvests through improving growth performance is an important goal of bivalve breeding. The identification of growth-related genes could benefit our understanding of the mechanism underlying growth regulation of bivalves and provide genomic targets for the breeding of fast-growth bivalves.

The somatostatin receptors (SSTRs) are negative regulators of growth that were initially found in vertebrates. They belong to the G protein-coupled receptor (GPCR) family, exhibiting a characteristic seven transmembrane (7TM) domains [1]. In humans, mutations in *SSTR* can trigger uncontrolled cell proliferation and ultimately lead to neuroendocrine tumors [2]. In *SSTR2* knockout mice, GH-mediated negative feedback was blocked and GH secretion was significantly increased [3]. In sheep, SNPs in *SSTR1* gene were significantly associated with body weight [4]. To date, a total of six *SSTR*s (*SSTR1~6*) have been identified in vertebrates, with *SSTR6* having been lost in mammals and only present in several fish species [5]. The lengths of the SSTR sequences range from 331 to 418 amino acids, exhibiting 40–60% structural similarity to each other. All SSTRs are activated by somatostatins, which possess two forms (somatostatin-14 and somatostatin-28). *SSTR4* causes prolonged activation of p38 MAPK, which in turn results in growth arrest [6], while *SSTR*s exert a negative effect on the release of growth hormone (GH), thereby inhibiting growth [7].

*SSTR*s were also found in invertebrates. In *Asterias rubens*, three *SSTRs* have been identified, and they are all activated by the ligand somatostatin 2, which is the homolog of somatostatin-14 [8]. In *Drosophila melanogaster*, two *SSTR* homologues were found to be expressed in the optic lobes of adult flies, an area devoted to the processing of visual information [9]. Previous studies showed that the loss of *SSTR*-2 in *Drosophila* could impair lipid and sugar mobilization, leading to hypoglycemia [10]. Moreover, in *Lacanobia oleracea*, *SSTR* inhibited larval growth via the suppression of gut myoactivity [11]. In *Rhodnius prolixus*, *SSTR* is expressed in the hindgut, midgut, and dorsal vessel. Furthermore, an investigation of SSTR signaling function revealed it roles in inhibiting post-prandial heart and gut activity [12]. In *Apis mellifera*, *SSTR* is mainly expressed in the brain, mostly in regions between the lobula, the medulla, and the lamina. Significant decreases in learning performance were observed in bees treated with the SSTR ligands, indicating that SSTR signaling may be involved in learning modulation [13]. However, SSTR studies in mollusks are relatively rare. In *Aplysia californica*, one *SSTR* gene was cloned and it could be activated by allatostatin C [14]. A study in venomous marine cone snails showed that SSTR was the target receptor of cone snail toxins, which is one of the most powerful venoms in nature [15]. However, whether *SSTR* functions in growth regulation in mollusks is still unknown.

The Yesso scallop *Patinopecten yessoensis* is an economically important bivalve in north-east Asia, including China and Japan. Since its introduction in the 1980s, *P. yessoensis* has become one of the most important cultivated mollusks in the northern Yellow Sea and Bohai Sea area of China [16]. According to the Fishery Statistical Yearbook of China, the yield of *P. yessoensis* reached 500,000 tons in 2023. In this study, we systemically identified the *SSTR* genes (*PySSTR*s) in the *P. yessoensis* genome and found that the expression of *PySSTR-1* was down-regulated in fast-growing scallops. We then verified the function of this gene through inhibiting the expression of its homolog in dwarf surf clams, *Mulinia lateralis*, a potential model organism for exploring bivalve gene functions [17,18,19]. Our findings indicate that the *SSTR* gene negatively regulates bivalve growth, implying its possible application in the genetic improvement of *P. yessoensis* to increase growth via gene editing.

## 2. Results

### 2.1. Characterization and Expression Analysis of PySSTRs

A total of seven *SSTRs* were identified in *P. yessoensis*, including *PySSTR-1*, *PySSTR-2*, *PySSTR-3*, *PySSTR-4*, *PySSTR-5*, *PySSTR-6*, and *PySSTR-7*, with the full-length sequences ranging from 546 bp to 107,352 bp. Their accession numbers are shown in Appendix A. A multiple-sequence alignment revealed that the amino acid sequences of the *PySSTR*s exhibited the typical characters of GPCRs, including seven transmembrane domains (Figure 1A). A phylogenetic analysis revealed two groups of SSTRs in mollusks, one grouping with the SSTRs from arthropods and annelids, and one that only contains mollusk SSTRs (Figure 1B). Three *PySSTR*s (*PySSTR-1*, *-2*, and *-3*) were expressed in adult tissues of scallops, and they were mainly expressed in the ganglia (Figure 1C). Among these three *PySSTR*s, *PySSTR-1* showed a significant difference in expression level between fast- and slow-growing scallops in both the PGCG (pedal and cerebral ganglia) and PVG (parietovisceral ganglia), and its down-regulation in fast-growing scallops implied a role in inhibiting scallop growth (Figure 1D).

### 2.2. Inhibition of MlSSTR-1 Expression Improved M. lateralis Growth

*MlSSTR-1* (accession number: PP663292.1) is the homolog of *PySSTR-1* (e-value = 5 × 10^−65^) in *M. lateralis* and it exhibited specific expression in the digestive gland (Figure 2A). A dsRNA targeting *MlSSTR-1* was produced using the *MlSSTR-1*-L4440 plasmid as the template (Appendix A). After a four-week period of RNAi, *MlSSTR-1* expression was significantly suppressed (*p* < 0.05, Figure 2B) in the *M. lateralis* digestive gland, and the body size of the RNAi group was larger than that of the control group (Figure 2C). The growth trait values of the RNAi group were significantly higher than those in the control group, with increases of 20%, 22%, 20%, 79%, and 92% in shell length, shell width, shell height, soft tissue weight, and muscle weight, respectively (Figure 2D, Table 1).

### 2.3. Transcriptome Analysis of the Digestive Gland of M. lateralis after RNAi

To understand the possible growth regulation pathways that *MlSSTR-1* is involved in, the transcriptome of digestive glands of *M. lateralis* with *MlSSTR-1* expression inhibition was analyzed. A total of 442 differentially expressed genes (DEGs) between the control group and RNAi group (log_2_|FC| > 1, *p* < 0.05) was obtained, among which, 237 were up-regulated and 205 were down-regulated in the RNAi group (Figure 3A). In the up-regulated genes in RNAi group, 64 genes were annotated, and the fatty acid-binding protein (*FABP*) was the most significant gene with a *p*-value of 1 × 10^−5^. To verify the correctness of the transcriptome analysis, we used qRT-PCR to test the expression changes of six randomly selected DEGs, including three significantly up-regulated genes and three significantly down-regulated genes (Figure 3B). The qRT-PCR results were consistent with those of the transcriptome analysis. We further explored the function of the DEGs through GO enrichment analysis. The results showed that 237 up-regulated DEGs were significantly enriched in 20 terms, and the most significant term was the cellular amino acid metabolic process, followed by fatty acid binding (Figure 3C). The KEGG enrichment analysis showed that the 237 up-regulated DEGs were enriched in 11 pathways, and the most significant pathway was the fat digestion and absorption pathway, followed by the insulin signaling pathway (Figure 3D).

## 3. Discussion

In this study, we characterized the *P. yessoensis SSTR* genes by analyzing their protein sequences and expression patterns, and then revealed the function of *MlSSTR-1* in growth regulation in the model bivalve *M. lateralis. P. yessoensis* possesses seven SSTRs, and all of them have 7TM domains, which are conserved to those in vertebrates [20], implying that SSTR may have similar roles across bivalves and vertebrates. Like all of the characterized vertebrate homologs [7], *PySSTR-1*, *-2*, and *-3* showed higher expression in the ganglia (Figure 1C), the nervous system of bivalves, but the other *PySSTR*s showed no expression in adult tissues. Moreover, *PySSTR-1* was also expressed in the kidney, the organ in which no SSTR expression was reported in vertebrates. The function of *PySSTR-1* in the kidney of scallop needs further exploration.

Previous studies reported that vertebrate *SSTR*s are highly associated with growth regulation via a negative effect on the release of growth hormone [4,21,22]. In invertebrates, SSTRs have been identified in echinoderms, arthropods, and mollusks. In echinoderms, three *SSTR*s were identified in echinoderms, arthropods, and mollusks [8,9,14]. In echinoderms, three *SSTR*s were identified in *A. rubens*, and only somatostatin 2 could act as the ligand of the three SSTRs, which regulate the relaxation of the tube feet and cardiac stomach preparations [8]. In arthropods, *SSTR* may suppress the larval growth of *L. oleracea* via inhibiting gut myoactivity [11]. Moreover, *SSTR* signaling in *R. prolixus* inhibited post-prandial heart and gut activity [12]. In *Homarus americanus*, *SSTR* is involved in regulating the cardiac neuromuscular system through consistently decreasing the frequency of heart contractions [23]. In mollusks, one SSTR was identified in *A. californica*, and it was demonstrated that allatostatin C was the ligand of the SSTR [14]. SSTR was also found to be the target receptor of toxins from venomous marine cone snails [15]. However, there are no reports on *SSTR*′s role in growth regulation in mollusks. Our data revealed that SSTR functions in growth regulation as an inhibitor. *PySSTR-1* expression was significantly down-regulated in fast-growing scallops, indicating that it may play a role in inhibiting *P. yessoensis* growth. Moreover, inhibiting *MlSSTR-1* expression resulted in significant enhancements in all the growth-related traits in *M. lateralis*, consist with the speculated role in *P. yessoensis*. These results suggested that SSTR may have a conserved function in growth regulation across vertebrates and bivalves. Meanwhile, we found that *MlSSTR-1* expression inhibition resulted in a 20–22% increase in shell size and 79% and 92% increase in soft tissue weight and muscle weight, respectively, implying that *MlSSTR-1* inhibition might have stronger effects on the growth of soft tissues than that of shells. Further in-depth studies on the dynamic relationship between *MlSSTR-1* expression and growth performance using RNAi and gene editing of *MlSSTR-1* could benefit our understanding of the mechanism underlying *MlSSTR-1*’s inhibition of clam growth.

*FABP* was the most significantly up-regulated gene after inhibiting *MlSSTR-1* expression. The *FABP* gene encodes a small-molecule protein involved in lipid absorption and utilization, and is closely linked to fatty acid levels [24]. Elevated expression of *FABP* may lead to fast growth in fish by accelerating the absorption and utilization of lipids [25]. However, the up-regulation of *FABP* expression may accelerate fatty acid metabolism, leading to the promotion of fat digestion and absorption [26]. Our results suggest that *MlSSTR-1* may inhibit *FABP* expression, thereby exerting a negative effect on the absorption and utilization of fatty acids, which, in turn, inhibits *M. lateralis* growth.

The insulin and IGF signaling pathways are crucial regulators of glucose, lipid, and protein metabolism [27,28]. In vertebrates, *SSTR* negatively regulates growth via the suppression of growth hormone and *IGF-1* [29,30]. Fish *SSTR-6* was shown to inhibit the expression of the *IGF* gene, which then reduced the speed of liver metabolism, thus inhibiting growth [22]. After inhibiting *MlSSTR-1* expression, *M. lateralis* growth was promoted, and multiple metabolic processes in the digestive gland were up-regulated. However, the insulin signaling pathway was significantly up-regulated but not the IGF expression level (Appendix A) or IGF signaling pathway. These findings indicate that *MlSSTR-1* may inhibit clam growth by regulating the metabolism of *M. lateralis* through the insulin signaling pathway, but not the related IGF pathway as in vertebrates. As *MlSSTR-1* was specifically expressed in the digestive gland of *M. lateralis*, further analyses on the temporal and spatial expression changes of this gene in digestive glands after RNAi could be helpful to understanding the mechanism of this gene in clam growth regulation.

In conclusion, we systemically identified the *SSTR gene*s in *P. yessoensis,* and verified that the *SSTR-1* gene functions in growth inhibition in bivalves. We observed that the *SSTR-1* gene was down-regulated in both fast-growing scallops and clams, and the growth traits of the clams were enhanced after *MlSSTR-1* expression suppression by RNAi. Furthermore, *SSTR-1* may regulate growth via inhibiting fat metabolism, amino acid metabolism, and the insulin signaling pathway. This study deepens our understanding of *SSTR* functions in invertebrates and provides an excellent gene for gene editing in bivalves to improve growth. Moreover, this study will be helpful in understanding the molecular mechanisms of bivalve growth regulation by somatostatin signaling.

## 4. Materials and Methods

### 4.1. Sample Collection

*P. yessoensis* was collected randomly from one cultured population in Dalian, and their growth traits, including shell length (SL), shell width (SW), shell height (SH), body weight (BW), soft tissue weight (STW), and muscle weight (MW), were analyzed by principal component analysis (PCA). According to the PC1 values (Appendix A), the top and bottom 10 individuals were designated as the fast- and slow-growing scallops, respectively.

To perform RNAi experiments, 60 two-month-old *M. lateralis* from one population were randomly divided into two groups: the control group (30 clams; SL: 5.36 ± 0.22 mm) and RNAi group (30 clams; SL: 5.45 ± 0.25 mm; *p* > 0.05). The groups were cultured in a 15 L glass tank with 10 L filter seawater at 21–22 °C. For each group, the seawater was replaced once a day and the *M. lateralis* were fed with *Chlorella pyrenoidesa* twice a day at 9:00 a.m. and 9:00 p.m., respectively, with a cell density of 4000 cells mL^−1^ every time.

### 4.2. Characterization and Expression Analysis of SSTRs in P. yessoensis

The SSTR sequences of interest (Appendix A) were obtained from NCBI (https://www.ncbi.nlm.nih.gov/ accessed on 1 September 2023). The sequences of *SSTR*s in *M. lateralis* were shown in Appendix A. The open reading frames (ORFs) of the *SSTR*s were obtained using ORF Finder (https://www.ncbi.nlm.nih.gov/orffinder/ accessed on 1 September 2023). We also aligned the *PySSTR* sequences with those of other species using Clustal W [31] and predicted transmembrane domains using the TMHMM Server v. 2.0 (http://www.cbs.dtu.dk/services/TMHMM/ accessed on 5 September 2023). The maximum likelihood (ML) phylogenetic tree was constructed using iQ-TREE [32], using the model of Q.pfam + F + R6. The expression levels of the *SSTR*s in *P. yessoensis* were analyzed based on transcriptomic data [33]. Transcripts per kilobase of exon model per million mapped read (TPM) values below two were considered to indicate no expression. The differences in *PySSTR*s expression levels between fast- and slow-growing scallops were detected by ANOVA at α = 0.05 using IBM SPSS Statistics 20.0 (SPSS, Chicago, IL, USA). A *p*-value < 0.05 was considered significant.

### 4.3. Inhibition of MlSSTR-1 Expression in M. lateralis

Total RNA was extracted from *M. lateralis* digestive glands (the only organ expressing *MlSSTR-1*) using the guanidinium isothiocyanate method [34] and then digested with DNase I (TaKaRa, Japan) to remove any residual DNA. The first-stand cDNA was synthesized using Reverse Transcriptase M-MLV (RNase H-) (TaKaRa, Shiga, Japan). The partial gene fragment of *MlSSTR-1* was amplified by PCR using the primers in Table 2 and subcloned into the L4440 vector. Competent cells (HT115), which were transformed with the *MlSSTR-1*-L4440, were used to produce double-stranded RNA (dsRNA) of *MlSSTR-1* [35].

To minimize the possibility of off-target effects, we submitted the sequence of *MlSSTR-1* gene to siDirect 2.0 (http://sidirect2.rnai.jp/ accessed on 25 May 2023) to predict and screen the RNAi target sequence. siDirect 2.0 is an online server that utilizes a fast and sensitive homology search algorithm to minimize any off-target effects and ensure functional dsRNA designs. In addition, the RNAi target sequence that was uniquely aligned with the *M. lateralis* genome was selected for dsRNA synthesis to further avoid off-target effects.

During the RNAi experiment, the control group of *M. lateralis* was fed with *C. pyrenoidesa* mixed with HT115 (DE3) containing the empty L4440 plasmid, and the RNAi group was fed with *C. pyrenoidesa* mixed with HT115 (DE3) containing the *MlSSTR-1*-L4440 plasmid. Specifically, HT115 bacteria containing the recombinant plasmid and empty plasmid were grown overnight with shaking in LB with ampicillin (50 μg mL^−1^) and tetracycline (12.5 μg mL^−1^) at 37 °C. One milliliter of overnight culture was diluted 100-fold in 100 mL of fresh LB medium containing ampicillin (50 μg mL^−1^) and tetracycline (12.5 μg mL^−1^). Before feeding, 0.5 mM IPTG was added to the culture of HT115 (DE3) cells (with OD_600nm_ reaching 0.4–0.6) for the two groups for 4 h at 37 °C to induce the transcription of *MlSSTR-1* dsRNA. Then, the induced bacterial cultures were centrifuged and the bacterial pellets were resuspended in 20 mL *C. pyrenoidesa* algae culture liquid. Algae/bacteria co-inoculum was produced by mixing the algal culture and bacterial suspension at a ratio of 10 bacterial cells per algal cell, with a final *C. pyrenoidesa* concentration of 4000 cells mL^−1^. Food reserves were renewed with fresh algae/bacteria co-inoculum every 24 h. The bacteria could be either attached to the algae or free living [36]. The bacterial adsorption rate on the algae was calculated as 99%, according to a previous study in *Crassostrea gigas* [37].

After RNAi, the digestive glands of all *M. lateralis* were dissected, frozen in liquid nitrogen, and stored at −80 °C. We set up five biological replicates, each containing a mixture of digestive glands collected from six individuals. Quantitative real-time PCR (qRT-PCR) was used to determine the expression level of *MlSSTR-1* in the digestive gland of *M. lateralis*. Total RNA extraction and cDNA synthesis were performed using the method described above. The cDNA was used as the template for qRT-PCR and elongation factor 1 alpha (*ef1α*) was chosen as the reference gene [38]. The qRT-PCR program was run on a LightCycler480^®^ II (Roche, Indianapolis, IN, USA) and each sample was run in triplicate. At the end of each PCR reaction, a dissociation analysis was used to confirm the amplification specificity. The data were analyzed by the 2^−∆∆Ct^ method based on the CT values of *MlSSTR-1* and *ef1a* to calculate the relative mRNA expression level [39,40]. The differences in *MlSSTR-1* expression levels were detected by ANOVA at α = 0.05 using IBM SPSS Statistics 20.0 (SPSS, Chicago, IL, USA). A *p*-value < 0.05 was considered significant.

### 4.4. Analysis of Growth Changes after Inhibiting MlSSTR-1 Expression

After four weeks of *MlSSTR-1* expression inhibition, the shell length (SL, mm), shell width (SW, mm), shell height (SH, mm), body weight (BW, g), soft tissue weight (STW, g), and muscle weight (MW, g) of *M. lateralis* were measured. A Gauss distribution analysis showed that the skewness of shell length, shell width, shell height, soft tissue weight, and muscle weight of *M. lateralis* in the control group were −0.561, −0.035, −0.495, 0.209, and 3.766, respectively. The skewness of shell length, shell width, shell height, soft tissue weight, and muscle weight of *M. lateralis* in the RNAi group were −0.395, −0.378, −0.221, 1.864, and 4.854, respectively. Thus, the significant differences in shell length, shell width, and shell height between the two groups were detected by Student’s *t*-test, assuming equal variance, or chi-square analysis. The significant differences in soft tissue weight and muscle weight were detected by ANOVA at α = 0.05. For all statistical results, a probability of *p* < 0.05 was considered significant. The analyses were performed with IBM SPSS Statistics 20.0 (SPSS, Chicago, IL, USA).

### 4.5. Transcriptome Analysis of the Digestive Gland of M. lateralis after RNAi

Total RNA from *M. lateralis* digestive glands was used to construct the RNA-seq library using the VAHTS Universal V8 RNA-seq Library Prep Kit (Vazyme, Nanjing, China). The concentration of the library was determined by a Qubit RNA Assay Kit (Invitrogen, Carlsbad, CA, USA). The prepared libraries were subjected to paired-end 150 bp (PE150) sequencing on the Illumina HiSeq 2500 platform.

The high-quality reads of each sample were aligned to the reference genome of *M. lateralis* [17,18]. Read counts mapped to each gene were obtained using HTSeq [28]. TPM values based on read counts and transcript lengths were used to evaluate the expression level of each gene. The gene expression profiles of the digestive glands of the RNAi group and the control group were compared. The differentially expressed genes (DEGs) were obtained using the Bioconductor package edgeR (v3.6.1) in R language [41], using the threshold log_2_|FC| > 1 and *p* < 0.05. For the Gene Ontology (GO) [42] and the Kyoto Encyclopedia of Genes and Genomes (KEGG) [43] analyses, the enriched GO terms (GO level = 4) and KEGG pathways of the DEGs were analyzed by the Enrich Pipeline.

To verify the transcriptome results, six genes were randomly selected, including three significantly up-regulated genes (*Fmo5*, *RTase* and *C4ST1*) and three significantly down-regulated genes (*CHRNA4*, *GDH* and *OSF-2*). The expression levels of these genes in digestive glands were detected by qRT-PCR, as described in Section 4.4. Each sample was run in triplicate, with five samples for each gene. The primers used in the verification are shown in Table 2.

## Figures and Tables

**Figure 1 ijms-25-04813-f001:**
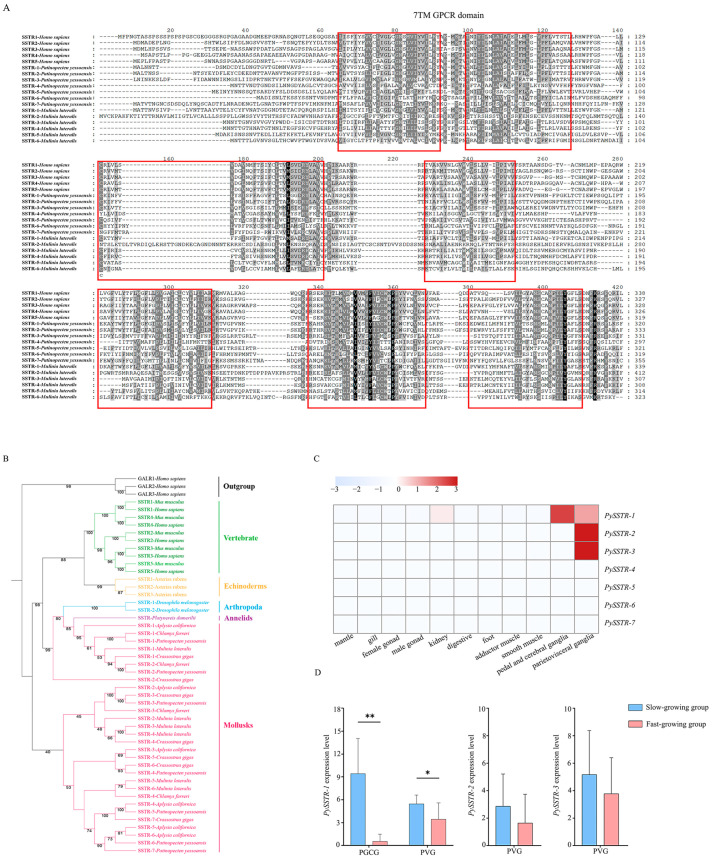
(**A**) Alignment of amino acid sequences of SSTRs. The black block represents a position where the amino acid is completely conserved across all sequences, denoting a highly conserved region. The dark gray block represents a position where the amino acid is similar in most sequences, indicating a moderate level of conservation. The light gray block indicates a position where the amino acid is mostly similar, but there are some variations. The red boxes indicate the position of seven transmembrane domains. The numbers above the image represent the amount of amino acids in the sequences. (**B**) The phylogenetic tree of SSTRs in bilaterians with galanin-type receptors (GALRs) as the outgroup. (**C**) *PySSTR* expression levels in adult tissues. The expression levels, as represented by TPM values, are shown in the gradient heat map with colors ranging from white (low expression) to red (high expression). (**D**) Differences in expression of *PySSTR*s between fast- and slow-growing scallops. * represents *p* < 0.05 and ** represents *p* < 0.01.

**Figure 2 ijms-25-04813-f002:**
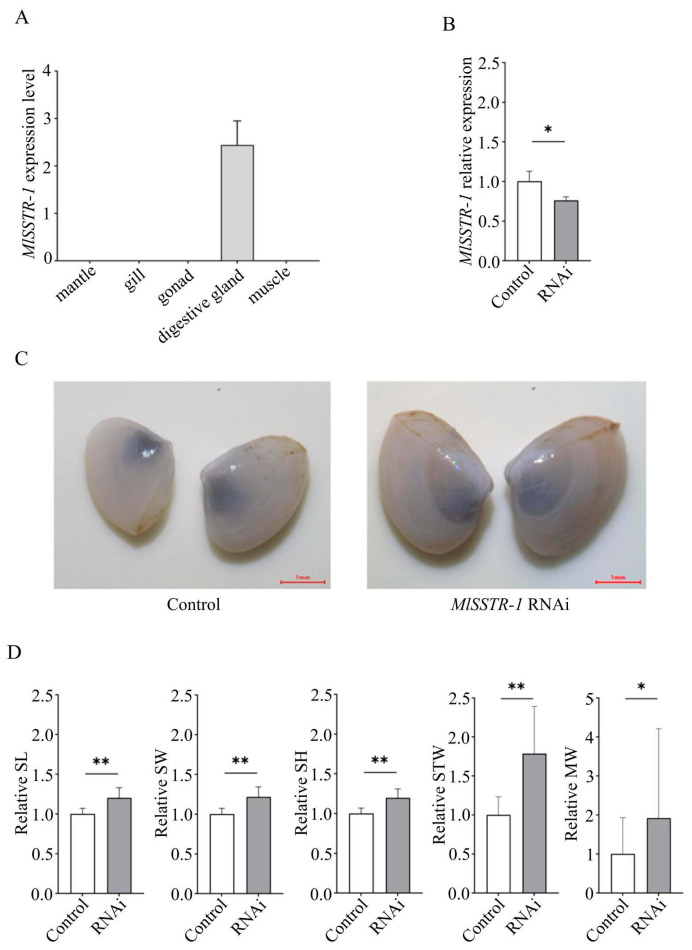
(**A**) Expression pattern of *MlSSTR-1* in adult tissues of *M. lateralis*. (**B**) Relative expression level of *MlSSTR-1* in digestive glands after RNAi. (**C**) Body size changes of *M. lateralis* after *MlSSTR-1* expression suppression. (**D**) Comparison of relative growth trait values of shell length (SL), shell width (SW), shell height (SH), soft tissue weight (STW), and muscle weight (MW) between control and RNAi groups after four-week inhibition of *MlSSTR-1* expression. * represents *p* < 0.05 and ** represents *p* < 0.01.

**Figure 3 ijms-25-04813-f003:**
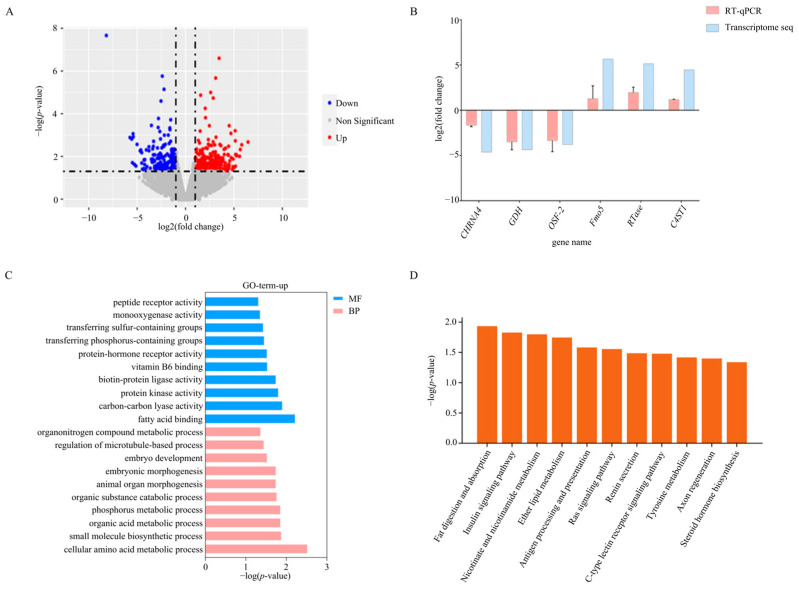
(**A**) The volcano plot of the DEGs in the digestive gland after *MlSSTR-1* expression inhibition. (**B**) Verification of the expression changes of six DEGs from the transcriptome analysis by qRT-PCR. (**C**) GO enrichment and (**D**) KEGG analysis of up-regulated DEGs in the digestive gland after the inhibition of *MlSSTR-1* expression.

**Table 1 ijms-25-04813-t001:** Changes in growth-related traits of *M. lateralis* after inhibiting *MlSSTR-1* expression.

Growth-Related Trait	Control	RNAi
Shell length (mm)	7.30 ± 0.52	8.78 ± 0.92 (**)
Shell width (mm)	3.04 ± 0.22	3.70 ± 0.38 (**)
Shell height (mm)	5.85 ± 0.41	7.01 ± 0.66 (**)
Body weight (g)	0.0720 ± 0.0148	0.1247 ± 0.0329 (**)
Soft tissue weight (g)	0.0422 ± 0.0098	0.0753 ± 0.0255 (**)
Muscle weight (g)	0.0052 ± 0.0049	0.0101 ± 0.0120 (**)

Notes: **, *p* < 0.01. Data are shown as mean ± SD.

**Table 2 ijms-25-04813-t002:** Primers used in the study.

Name	Primer Sequence
*MlSSTR-1*-RNAi-F	CCGCGGATTGGTGTTTCCAGGTCGAACTTAC
*MlSSTR-1*-RNAi-R	CTCGAGCGGCTCTTCGTTACCATTCATTT
*MlSSTR-1*-qPCR-F	GCGAAAGGTTTGAGTTCACATCTAT
*MlSSTR-1*-qPCR-R	CCAAGCATCGGCTCTTCGTTA
*ef1a*-qPCR-F	CAGCACTGAACCACCATACA
*ef1a*-qPCR-R	CAGCCTGAGATTGGCACGAA
*MlFmo5*-qPCR-F	CGGGAGTTGGGATGTGACGGTT
*MlFmo5*-qPCR-R	GTGAATGTAGCGTCGTGCCCTGG
*MlRTase*-qPCR-F	TAGACTTGATAACTCGACAAGAGCCA
*MlRTase*-qPCR-R	CGGTGAGATGATAATCGTAACTGTGA
*MlC4ST1*-qPCR-F	GTATGGCGGAAACTTCAAATGC
*MlC4ST1*-qPCR-R	TGGTCTGGTTATTGTCTGGGATT
*MlTcb1*-qPCR-F	CTGTCGTCGTTCCTTACCTCAAT
*MlTcb1*-qPCR-R	TACTCGTCTGTCCAACAAATCCC
*MlCHRNA4*-qPCR-F	ATGTAGTGCGTGCTCATGGAGTG
*MlCHRNA4*-qPCR-R	CAATGGCGAGGAGTAAGGTGATA
*MlGDH*-qPCR-F	AGGGTTTGCGAGTGGTGGATGC
*MlGDH*-qPCR-R	CCGCGAATCATGTCTGCTGCC
*MlOSF-2*-qPCR-F	GAACTCCTGGCTGCGGACCCT
*MlOSF-2*-qPCR-R	TGGCAGCTGGTAGCGCAGAGAA

## Data Availability

Data are contained within the article and Appendix A.

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
