# Peer review of "Somatostatin Receptor Gene Functions in Growth Regulation in Bivalve Scallop and Clam"

_ijms, 2024, doi:10.3390/ijms25094813_

Round 1
Reviewer 1 Report
Comments and Suggestions for Authors
The manuscript entitled "Somatostatin Receptor gene functions in Growth Regulation in Marine Bivalve" delves into the genetic underpinnings of growth regulation in Patinopecten yessoensis, utilizing a compelling approach that combines transcriptomic analysis and RNAi techniques applied in Mulinia lateralis. The identification and functional analysis of SSTR genes offer novel insights with practical implications in the field of aquaculture. The work is well presented: abstract, introduction, discussion, methods and conclusions are coherent. The methodology employed is adequate for the aims and the results obtained seem to be promising, with the authors discussing the putative mechanisms of action involved in the reduction of the side effects produced by melittin in the studied conditions. The conclusions are well supported by the methods, results, and the proposed mechanisms by the authors. Indeed, the identification of seven SSTRs in P. yessoensis is a significant contribution. While the manuscript is informative and presents a potentially significant finding, several improvements are necessary to enhance the clarity, completeness, and impact of the study.
Abstract: The abstract succinctly introduces the topic, highlighting the investigation of somatostatin receptors in bivalve growth. However, the presentation could be enhanced by delineating the novelty of the RNAi method applied. Specific points of attention include:
- Clarify the role of RNAi in the methodological framework and its relevance in the context of the findings.
- Clearly state the main conclusions and their potential impact on bivalve aquaculture to underscore the practical importance of the study.
Introduction: The introduction is well-composed but could be strengthened by addressing the following:
- A broader literature review concerning the function of SSTRs in invertebrates would be beneficial. This would place the study in a wider evolutionary context, potentially increasing its interest to a broader audience.
Materials and Methods:
the information provided establishes a good foundation. However, to further enhance the reproducibility of your work, it would be beneficial to expand upon the existing descriptions. Specifically:
- Elaborate on the control treatments within the RNAi experiments, clarifying any variations in dosages, timings, and delivery methods used. This additional detail will aid other researchers in mirroring your experimental conditions as closely as possible.
- Regarding statistical analyses, while the chosen tests and significance levels are noted, providing a more in-depth explanation of the decision-making process behind selecting these methods would be advantageous. If applicable, include any software names and version numbers, parameter settings, and justifications for the chosen statistical approach. This level of detail is invaluable for ensuring that others can accurately evaluate and reproduce your results.
Results:
- There is a need for a more integrated presentation of gene expression data with growth outcomes. Figures need improvement for clarity and resolution to ensure that the presented data are interpretable and meet publication standards (i.e., Figure 1).
Discussion:
- Expand the comparison of the observed SSTR functions with those reported in other species to better articulate the significance of your findings.
- Address potential off-target effects of RNAi, which could contribute to the observed phenotypes. Discussion of these aspects would provide a more balanced view of the results.
Conclusions:
- The conclusions would benefit from a more pronounced emphasis on the implications of the findings for the marine biology and aquaculture fields, along with a clearer statement of the future directions suggested by this work.
Reviewer 2 Report
Comments and Suggestions for Authors
The article “Somatostatin Receptor gene functions in Growth Regulation in Marine Bivalve” by Zhang et al. is dealing with a very interesting topic, which reveals the growth-inhibitory role of SSTR-1 in bivalve. Somatostatin receptor (SSTR) is a conserved negative regulator of growth in vertebrates. In this study, they identified seven SSTRs (PySSTRs) in Yesso scallop, Patinopecten yessoensis. Among the three PySSTRs (PySSTR-1, -2 and -3) expressed in adult tissues, PySSTR-1 showed significantly lower expression in fast-growing scallops than in slow-growing scallops. Then the function of this gene in growth regulation was evaluated with RNAi in dwarf surf clam (Mulinia lateralis), which resulted in significant increase of shell length, shell width, shell 26 height, soft tissue weight and muscle weight by 20%, 22%, 20%, 79%, and 92%, respectively. Transcriptome analysis indicated that, genes in fat digestion and absorption pathway and the insulin pathway are significantly enriched in the up-regulated genes after MlSSTR-1 inhibition. However, some major issues mentioned seriously affect the quality of this MS.
1. Add some background on P. yessoensis farming in line 61, such as production, production areas, etc.
2. Please add phylogenetic tree.
3. Please provide accession numbers of relevant genes such as pyPySSTR-1, etc.
4. What's the ethics authorization number?
5. Please provide all primer sequences used in this experiment.
6. It is necessary to randomly select some genes to verify the correctness of the transcriptome! Please supplement.
7. please provide information about tests by which Gauss distribution has been evaluated in line 228.
8. I'm confused about the concentration of dsRNA used in line 209, although you describe the preparation of double-stranded RNA in 4.3.
9. I want to know how shellfish filter pelleted feeds. Also, as seen in the result of Figure 2-B, your interference efficiency is very low, about 25% only.
Comments on the Quality of English LanguageThe English needs a complete editing by a professional English editor.
Round 2
Reviewer 2 Report
Comments and Suggestions for Authors
The authors do not clearly explain why RNAi has such a high phenotype effect but such a low interference efficiency. Although the phenotypic effect of RNA interference produced by the authors is significant, the authors' method of evaluating the efficiency of RNA interference is wrong. Firstly, the temporal and spatial expression pattern of this gene in digestive gland was not determined. The evaluation of RNAi efficiency was based on the assumption that the expression of the gene remained unchanged within four weeks and that both the experimental group and the control group were in the same expression period, but there was no evidence for this assumption. Efficiency is not proven by data from one test site alone. It is suggested that the author re-conduct the RNAi efficiency evaluation test or delete this part.
Comments on the Quality of English LanguageExtensive editing of English language required
Round 3
Reviewer 2 Report
Comments and Suggestions for Authors
The English needs a editing by a professional English editor.
Comments on the Quality of English LanguageThe English needs a editing by a professional English editor.